

# Light Nuclei Production in Au+Au Collisions at $\sqrt{s_{\mathrm{NN}}} = 3$ GeV from the STAR experiment

**Hui Liu$^{1\star}$**

**1** Key Laboratory of Quark & Lepton Physics (MOE) and Institute of Particle Physics, Central China Normal University, Wuhan 430079, China

$\star$ huihuiliu@mails.ccnu.edu.cn

## Abstract

**Light nuclei production is expected to be sensitive to baryon density fluctuations and can be used to probe the signatures of QCD critical point and/or a first-order phase transition in heavy-ion collisions. In this proceedings, we present the spectra and yields of protons ($p$) and light nuclei ($d$, $t$, $^{3}$He, $^{4}$He) in Au+Au collisions at $\sqrt{s_{\mathrm{NN}}} = 3$ GeV by the STAR experiment. Finally, it is found that the kinetic freeze-out dynamics (temperature $T_{kin}$ $vs.$ average radial flow velocity $\langle \beta_T \rangle$) at $\sqrt{s_{\mathrm{NN}}} = 3$ GeV extracted with the blast-wave model deviate from the trends at high energies ($\sqrt{s_{\mathrm{NN}}} = 7.7$ - 200 GeV), indicating a different medium equation of state.**

## 1 Introduction

The Beam Energy Scan (BES) program at the Relativistic Heavy-ion Collider (RHIC) aims at understanding the phase structure and properties of strongly interacting matter under extreme conditions. In particular, it was proposed to search for a possible phase boundary and critical point (CP) of the phase transition from hadron gas to quark-gluon plasma (QGP) [1].

Light nuclei production is sensitive to the baryon density fluctuations and can be used to probe the QCD phase transition in relativistic heavy-ion collisions [2]. At RHIC BES-I energies, the STAR experiment has collected data from Au+Au collisions at $\sqrt{s_{\mathrm{NN}}} = 7.7$, 11.5, 14.5, 19.6, 27, 39, 54.4, 62.4, and 200 GeV and measured the production of light nuclei (deuteron and triton) [3]. In this proceedings, the transverse momentum spectra of proton ($p$), deuteron ($d$), triton ($t$), $^{3}$He, and $^{4}$He in Au+Au collisions at $\sqrt{s_{\mathrm{NN}}} = 3$ GeV measured at various rapidity ranges are presented. In addition, we show the rapidity and centrality dependence of dN/dy and $\langle p_T \rangle$. Finally, we discuss the kinetic freeze-out temperature $T_{\mathrm{kin}}$ and average radial flow velocity $\langle \beta_T \rangle$.

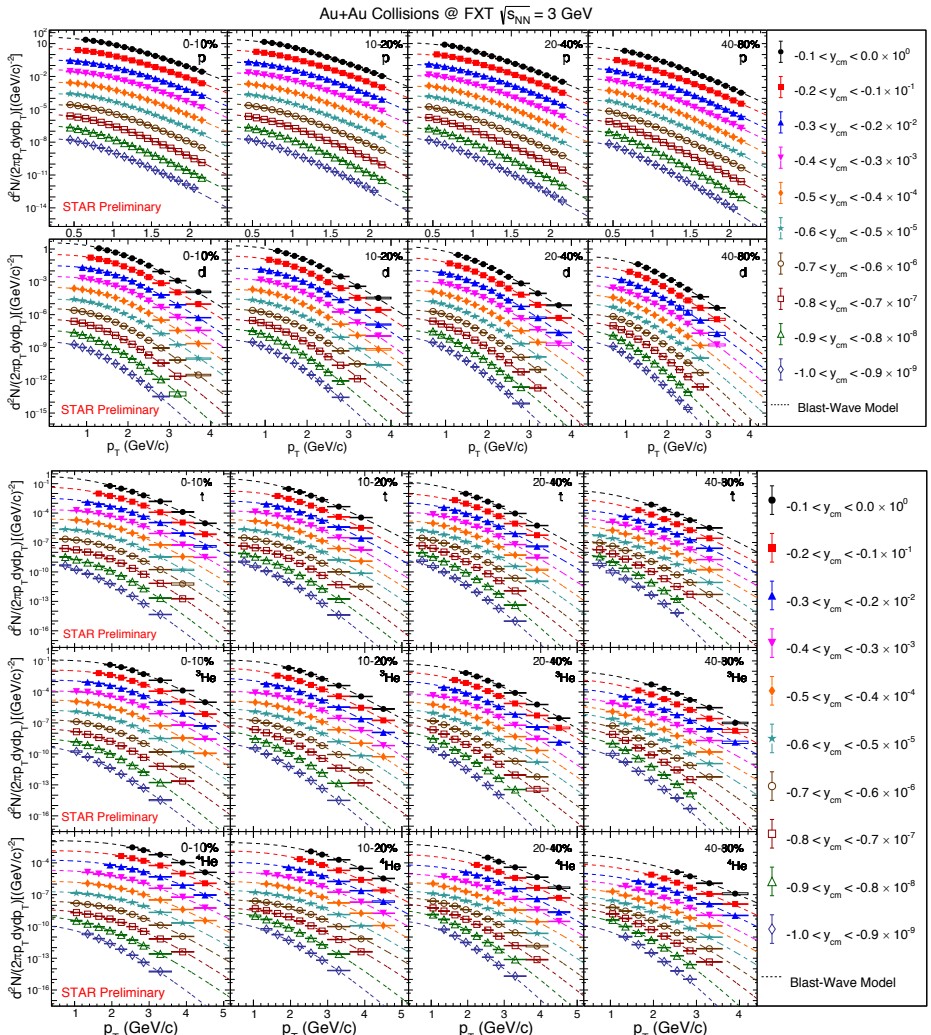

Figure 1: Transverse momentum spectra for proton, deuteron, triton, $^3$He, and $^4$He in Au+Au collisions at $\sqrt{s_{NN}} = 3$ GeV. The dashed lines are fit by blast-wave model.

## 2 Experiment and Analysis Details

In 2018, RHIC started the second phase of the beam energy scan program (BES-II). The STAR Fixed-Target (FXT) program was proposed to achieve lower center-of-mass energies and higher baryon density regions. The target was installed in the vacuum pipe at 200 cm to the west of the nominal interaction point of the STAR detector.

The dataset used in this analysis is obtained from the FXT program of Au+Au collisions at $\sqrt{s_{NN}} = 3$ GeV by the STAR expriment. Particle identification is done with two types of detectors: at low momentum by ionization energy loss (dE/dx) information from the Time Projection Chamber (TPC) and at high momentum by $m^2$ information from the Time of Flight (TOF). The total number of minimum bias triggered events used in this analysis is about 260 million.

The center-of-mass rapidity coverage for the FXT Au+Au collisions at $\sqrt{s_{NN}} = 3$ GeV is from -1.0 to 0.2. The rapidity range of each particle (-1.0 to 0 in this analysis) was partitioned into 10 uniform intervals of bin width 0.1. The centralities are divided into 0-10%, 10-20%, 20-40%, and 40-80%, respectively.

For the final results, the several corrections must be applied. Due to the limited detector

acceptance and efficiency, the raw spectra are corrected with TPC tracking efficiency and TOF matching efficiency, and the energy loss correction is also applied due to the loss of energy when particles traversing the detector material.

As the weak decay feed-down contribution from strange baryons to the proton is below 2% at this energy, we didn't apply feed-down correction to the inclusive proton yield. On the other hand, due to lack of anti-particle production, the subtraction of possible background particles knocked-out from beam-pipe is not done in this analysis.

# 3 Results

Figure 1 shows the $p_T$ spectra for proton, deuteron, triton, $^3$He and $^4$He in Au+Au collisions at $\sqrt{s_{NN}} = 3$ GeV. For illustration purpose, different rapidly slices are scaled by different factors. The dashed lines are fit by the blast-wave model.

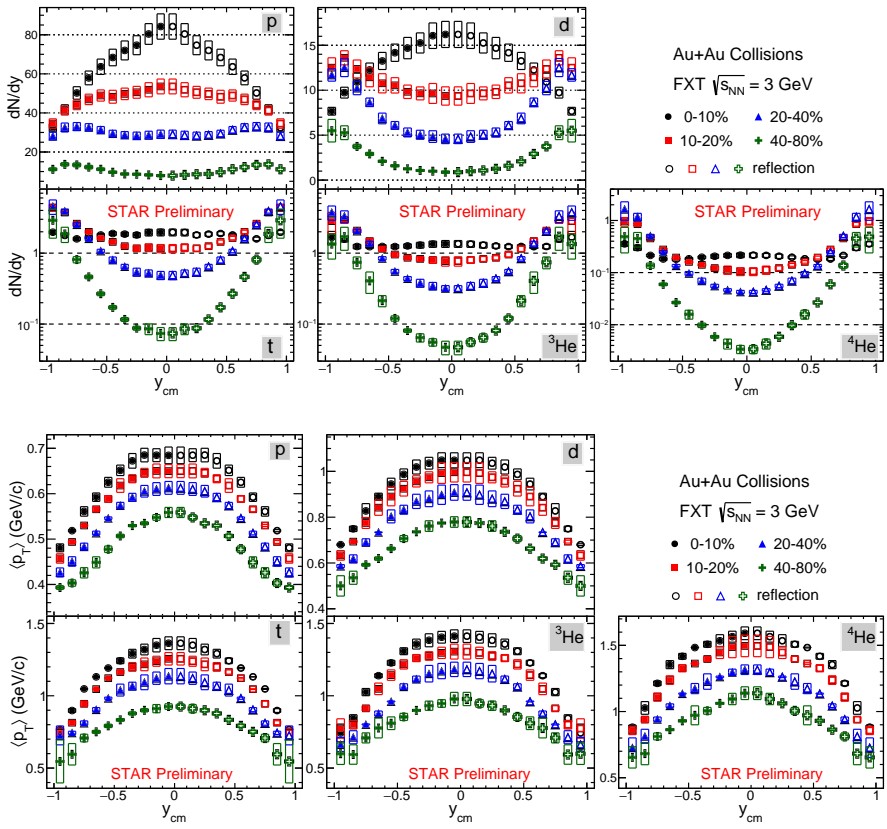

Figure 2: (top) dN/dy and (bottom) $\langle p_T \rangle$ distribution of proton, deuteron, triton, $^3$He, and $^4$He in Au+Au collisions at $\sqrt{s_{NN}} = 3$ GeV. Solid markers obtained by real data, open markers are reflected by measured ranges. The boxes indicate the systematical uncertainties.

The blast-wave model function is given by [4]:

$$\frac{1}{2\pi p_T}\frac{d^2N}{dp_T dy} \propto \int_0^R r\,dr\,m_T I_0\left(\frac{p_T \sinh\rho(r)}{T_{kin}}\right)K_1\left(\frac{m_T \cosh\rho(r)}{T_{kin}}\right), \tag{1}$$

where $m_T$ is the transverse mass of particle, $I_0$ and $K_1$ are the modified Bessel functions, and $\rho(r) = \tanh^{-1}\beta_T$. The radial flow velocity $\beta_T$ in the region $0 \leq r \leq R$ can be expressed as

$\beta_T = \beta_S (r/R)^n$, where $\beta_S$ is the surface velocity, and $n$ reflects the form of the flow velocity profile (fixed $n = 1$ in this analysis). $\langle \beta_T \rangle$ can be obtained from $\langle \beta_T \rangle = \frac{2}{2+n} \beta_S$. The temperature $T_{kin}$ is a free parameter that can be extracted from the fit.

Figure 2 shows the rapidity dependence of dN/dy and $\langle p_T \rangle$ at different centralities. In a statistical approach to the formation of light nuclei, the yield is proportional to the spin degeneracy factor (2J+1), so one needs to divide the yield by the factor to get mass dependence [5–7]. Figure 3 left shows dN/dy as a function of particle mass for 0-10% central collisions, which shows an exponential decreased trend. Figure 3 right shows $\langle p_T \rangle$ as a function of particle mass, where the linear trend reflects the collective motion of light nuclei.

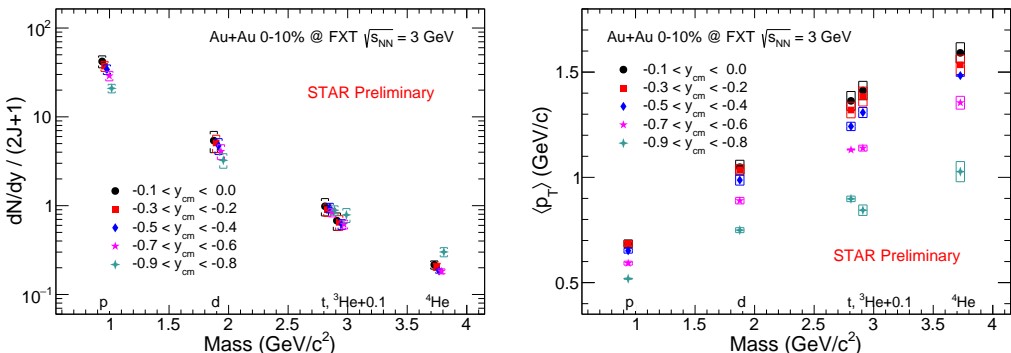

Figure 3: (left) dN/dy as an exponential function of particle mass at 0-10% central collisions from different rapidity windows, (right) $\langle p_T \rangle$ as a linear function of particle mass at 0-10% central collisions from different rapidity windows within the uncertainty.

The transverse momentum distributions of the different particles reflect the collective motion and bulk properties of the matter at kinetic freezeout [8], as Fig. 3 shows.

We fit the $p_T$ spectra of $\pi^{\pm}$, $K^{\pm}$, $p$ and light nuclei ($d$, $t$, $^3$He and $^4$He) simultaneously with Eq. 1 to obtain a common kinetic freeze-out temperature $T_{\text{kin}}$ and average radial flow velocity $\langle \beta_T \rangle$ at each centrality at $\sqrt{s_{\text{NN}}} = 3$ GeV. We also calculate the common parameters of $\pi^{\pm}$, $K^{\pm}$, $p$, $\bar{p}$, $d$ and $t$ measured at BES-I program [3,9,10]. Figure 4 shows $T_{\text{kin}}$ $vs.$ $\langle \beta_T \rangle$ distribution, the plotted total uncertainties are the quadratic sums of the statistical and systematic uncertainties, where the systematic uncertainty comes from the following three sources: 1) Fit different $p_T$ ranges; 2) Simultaneous fitting of different particle combination; 3) The blast-wave parameter $n$ being free or fixed to unity. Interestingly, we find the results from $\sqrt{s_{\text{NN}}} = 3$ GeV show a different trend comparing to those from BES-I energies. This indicates a different equation of state (EoS) of the medium created in Au+Au collisions at $\sqrt{s_{\text{NN}}} = 3$ GeV within the blast-wave model framework.

## 4 Conclusion

We report the measurements of the proton and light nuclei ($d$, $t$, $^3$He, and $^4$He) production in Au+Au collisions at $\sqrt{s_{\text{NN}}} = 3$ GeV from the STAR experiment. The $p_T$ spectra, dN/dy and $\langle p_T \rangle$ distributions with various rapidity windows at 0-10%, 10-20%, 20-40% and 40-80% centrality are presented.

Furthermore, an intriguing finding based on the blast-wave model is that we have observed that the distribution of $T_{\text{kin}}$ $vs.$ $\langle \beta_T \rangle$ at $\sqrt{s_{\text{NN}}} = 3$ GeV exhibits a completely different trend compared to high energies. These results reflect the different bulk properties at kinetic freezeout,

implying a different medium equation of state (EoS) at $\sqrt{s_{NN}} = 3$ GeV. With the upgrade of the STAR detector, high statistics data of Au+Au collisions have been collected from the BES-II and Fixed-Target programs, which will allow us to perform more precise measurements at lower energies.

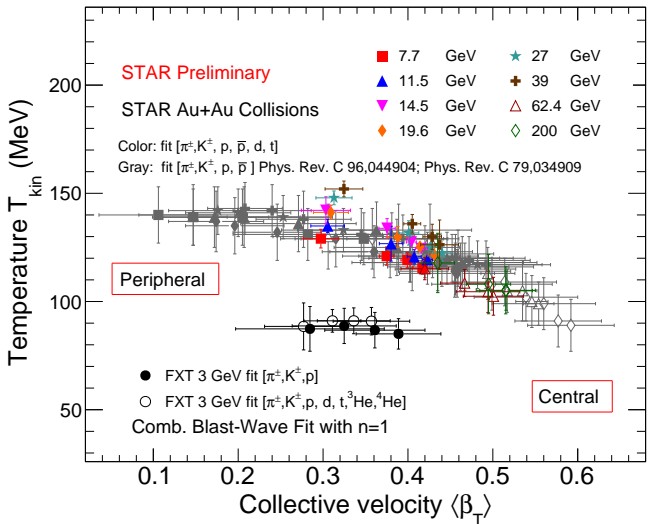

Figure 4: $T_{kin}$ *vs.* $\langle \beta_T \rangle$ distribution in Au+Au collisions at $\sqrt{s_{NN}} = 3$, 7.7, 11.5, 14.5, 19.6, 27, 39, 62.4, and 200 GeV, with the colourful points resulting from fits of BES-I data. Open and filled circles indicate different combinations of particles from the data at $\sqrt{s_{NN}} = 3$ GeV, the error bar contains statistical error and systematical uncertainty.

# Acknowledgements

This work is supported by the National Key Research and Development Program of China (Grants No. 2020YFE0202002 and No. 2018YFE0205201), the National Natural Science Foundation of China (Grants No. 12122505, No. 11890711 and No. 11861131009). And the Ministry of Science and Technology (MoST) under grant No. 2016YFE0104800 are also acknowledged.

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
