# Peer review of "Light Nuclei Production in Au+Au Collisions at $\sqrt{s_{\mathrm{NN}}}$ = 3 GeV from the STAR experiment"

_SciPost Physics Proceedings, doi:SciPost Phys. Proc. 10, 040 (2022)_

## Round 1 · Referee Report · Anonymous (Referee 1) · 2022-2-8

Strengths

This paper nicely describes the measurements of particles in Au-Au collisions at RHIC.
The measurements are presented and interpreted with state of the art models, possibly indicating a collision energy dependence.

Weaknesses

Surely due to the limited length of the paper the analysis part could profit from some small extension, discussing e.g. the contribution of the feed-down production to the final measurement.

Report

This article is worth of being published in this Journal

Requested changes

I could submit the following suggestions: - "restricted volume of phase space" it is unclear what the author is referring to, either specify more either drop the sentence - If possible in the data analysis section I would add a sentence discussing the contribution to measured yields of the particles produced in the material

  • validity: high
  • significance: high
  • originality: high
  • clarity: top
  • formatting: perfect
  • grammar: excellent

Author:  Hui Liu  on 2022-03-13  [id 2286]

(in reply to Report 1 on 2022-02-08)
Category:
question
answer to question
correction

Thank you for your comments.
Q1:the contribution of the feed-down production to the final measurement.
A1. Actually, for the number of secondary proton come from the strange baryons weak decay, it comes mainly from Lambda weak decay. At this energy, the ratio of Lambda/proton~2.5% (please find the proton and Lambda spectra on attachment), thus to consider the weak decay feed-down contribution from strange baryons to the proton is below 2%, in this proceeding, we have not applied the feed-down correction to the proton measurement.
Q2:"restricted volume of phase space" it is unclear what the author is referring to, either specify more either drop the sentence
A2: Done. Thank you!
Q3: If possible in the data analysis section I would add a sentence discussing the contribution to measured yields of the particles produced in the material
A3. At this energy, without sufficient antiparticles, so we also not removed the particles from material knock-out. Those corrections will be study in the future work.

Attachment:

Lambda_pro_0010_midY.pdf

---

## Round 3 · Author Response

Weaknesses: Surely due to the limited length of the paper the analysis part could profit from some small extension, discussing e.g. the contribution of the feed-down production to the final measurement.

Requested changes: I could submit the following suggestions: - "restricted volume of phase space" it is unclear what the author is referring to, either specify more either drop the sentence - If possible in the data analysis section I would add a sentence discussing the contribution to measured yields of the particles produced in the material

---

## Round 3 · List of Changes

-> Section2 Experiment and Analysis Details
We have added two paragraphs as the following:
For the final results, the several corrections must be applied. Due to the limited detector acceptance and efficiency, the raw spectra are corrected with TPC tracking efficiency and TOF matching efficiency, and the energy loss correction is also applied due to the loss of energy when particles traversing the detector material.
On the other hand, to consider the weak decay feed-down contribution from strange baryons to the proton is below 2%, in this proceeding, we have not applied the feed-down correction to the proton measurement. At this energy, without enough antiparticles, we also not removed the particles from material knock-out. Those corrections will be done in the future analysis.

-> Since the Legend font in Figure 1 is too small, we also change the legend on fig1

---

## Editorial Decision

published